# Efficient Inference of Continuous Markov Random Fields with Polynomial Potentials

**Shenlong Wang**
University of Toronto
slwang@cs.toronto.edu

**Alexander G. Schwing**
University of Toronto
aschwing@cs.toronto.edu

**Raquel Urtasun**
University of Toronto
urtasun@cs.toronto.edu

## Abstract

In this paper, we prove that every multivariate polynomial with even degree can be decomposed into a sum of convex and concave polynomials. Motivated by this property, we exploit the concave-convex procedure to perform inference on continuous Markov random fields with polynomial potentials. In particular, we show that the concave-convex decomposition of polynomials can be expressed as a sum-of-squares optimization, which can be efficiently solved via semidefinite programing. We demonstrate the effectiveness of our approach in the context of 3D reconstruction, shape from shading and image denoising, and show that our method significantly outperforms existing techniques in terms of efficiency as well as quality of the retrieved solution.

## 1 Introduction

Graphical models are a convenient tool to illustrate the dependencies among a collection of random variables with potentially complex interactions. Their widespread use across domains from computer vision and natural language processing to computational biology underlines their applicability. Many algorithms have been proposed to retrieve the minimum energy configuration, *i.e.*, maximum a-posteriori (MAP) inference, when the graphical model describes energies or distributions defined on a discrete domain. Although this task is NP-hard in general, message passing algorithms [16] and graph-cuts [4] can be used to retrieve the global optimum when dealing with tree-structured models or binary Markov random fields composed out of sub-modular energy functions.

In contrast, graphical models with continuous random variables are much less well understood. A notable exception is Gaussian belief propagation [31], which retrieves the optimum when the potentials are Gaussian for arbitrary graphs under certain conditions of the underlying system. Inspired by discrete graphical models, message-passing algorithms based on discrete approximations in the form of particles [6, 17] or non-linear functions [27] have been developed for general potentials. They are, however, computationally expensive and do not perform well when compared to dedicated algorithms [20]. Fusion moves [11] are a possible alternative, but they rely on the generation of good proposals, a task that is often difficult in practice. Other related work focuses on representing relations on pairwise graphical models [24], or marginalization rather than MAP [13].

In this paper we study the case where the potentials are polynomial functions. This is a very general family of models as many applications such as collaborative filtering [8], surface reconstruction [5] and non-rigid registration [30] can be formulated in this way. Previous approaches rely on either polynomial equation system solvers [20], semi-definite programming relaxations [9, 15] or approximate message-passing algorithms [17, 27]. Unfortunately, existing methods either cannot cope with large-scale graphical models, and/or do not have global convergence guarantees.

In particular, we exploit the concave-convex procedure (CCCP) [33] to perform inference on continuous Markov random fields (MRFs) with polynomial potentials. Towards this goal, we first show that an arbitrary multivariate polynomial function can be decomposed into a sum of a convex and

a concave polynomial. Importantly, this decomposition can be expressed as a sum-of-squares optimization [10] over polynomial Hessians, which is efficiently solvable via semidefinite programming. Given the decomposition, our inference algorithm proceeds iteratively as follows: at each iteration we linearize the concave part and solve the resulting subproblem efficiently to optimality. Our algorithm inherits the global convergence property of CCCP [25].

We demonstrate the effectiveness of our approach in the context of 3D reconstruction, shape from shading and image denoising. Our method proves superior in terms of both computational cost and the energy of the solutions retrieved when compared to approaches such as dual decomposition [20], fusion moves [11] and particle belief propagation [6].

## 2 Graphical Models with Continuous Variables and Polynomial Functions

In this section we first review inference algorithms for graphical models with continuous random variables, as well as the concave-convex procedure. We then prove existence of a concave-convex decomposition for polynomials and provide a construction. Based on this decomposition and construction, we propose a novel inference algorithm for continuous MRFs with polynomial potentials.

### 2.1 Graphical Models with Polynomial Potentials

The MRFs we consider represent distributions defined over a continuous domain $\mathcal{X} = \prod_i \mathcal{X}_i$, which is a product-space assembled by continuous sub-spaces $\mathcal{X}_i \subset \mathbb{R}$. Let $\mathbf{x} \in \mathcal{X}$ be the output configuration of interest, *e.g.*, a 3D mesh or a denoised image. Note that each output configuration tuple $\mathbf{x} = (x_1, \cdots, x_n)$ subsumes a set of random variables. Graphical models describe the energy of the system as a sum of local scoring functions, *i.e.*, $f(\mathbf{x}) = \sum_{r \in \mathcal{R}} f_r(\mathbf{x}_r)$. Each local function $f_r(\mathbf{x}_r) : \mathcal{X}_r \to \mathbb{R}$ depends on a subset of variables $x_r = (x_i)_{i \in r}$ defined on a domain $\mathcal{X}_r \subseteq \mathcal{X}$, which is specified by the restriction often referred to as region $r \subseteq \{1, \ldots, n\}$, *i.e.*, $\mathcal{X}_r = \prod_{i \in r} \mathcal{X}_i$. We refer to $\mathcal{R}$ as the set of all restrictions required to compute the energy of the system.

We tackle the problem of maximum a-posteriori (MAP) inference, *i.e.*, we want to find the configuration $\mathbf{x}^*$ having the minimum energy. This is formally expressed as

$$\mathbf{x}^* = \arg\min_{\mathbf{x}} \sum_{r \in \mathcal{R}} f_r(\mathbf{x}_r). \tag{1}$$

Solving this program for general functions is hard. In this paper we focus on energies composed of polynomial functions. This is a fairly general case, as the energies employed in many applications obey this assumption. Furthermore, for well-behaved continuous non-polynomial functions (*e.g.*, $k$-th order differentiable) polynomial approximations could be used (*e.g.*, via a Taylor expansion). Let us define polynomials more formally:

**Definition 1.** *A $d$-degree multivariate polynomial $f(\mathbf{x}) : \mathbb{R}^n \to \mathbb{R}$ is a finite linear combination of monomials,* i.e.,

$$f(\mathbf{x}) = \sum_{m \in \mathcal{M}} c_m x_1^{m_1} x_2^{m_2} \cdots x_n^{m_n},$$

*where we let the coefficient $c_m \in \mathbb{R}$ and the tuple $m = (m_1, \ldots, m_n) \in \mathcal{M} \subseteq \mathbb{N}^n$ with $\sum_{i=1}^{n} m_i \leq d \ \forall m \in \mathcal{M}$. The set $\mathcal{M}$ subsumes all tuples relevant to define the function $f$.*

We are interested in minimizing Eq. (1) where the potential functions $f_r$ are polynomials with arbitrary degree. This is a difficult problem as polynomial functions are in general non-convex. Moreover, for many applications of interest we have to deal with a large number of variables, *e.g.*, more than 60,000 when reconstructing shape from shading of a $256 \times 256$ image. Optimal solutions exist under certain conditions when the potentials are Gaussian [31], *i.e.*, polynomials of degree 2. Message passing algorithms have not been very successful for general polynomials due to the fact that the messages are continuous functions. Discrete [6, 17] and non-parametric [27] approximations have been employed with limited success. Furthermore, polynomial system solvers [20], and moment-based methods [9] cannot scale up to such a large number of variables. Dual-decomposition provides a plausible approach for tackling large-scale problems by dividing the task into many small sub-problems [20]. However, solving a large number of smaller systems is still a bottleneck, and decoding the optimal solution from the sub-problems might be difficult. In contrast, we propose to use the Concave-Convex Procedure (CCCP) [33], which we now briefly review.

## 2.2 Inference via CCCP

CCCP is a majorization-minimization framework for optimizing non-convex functions that can be written as the sum of a convex and a concave part, *i.e.*, $f(\mathbf{x}) = f_{\text{vex}}(\mathbf{x}) + f_{\text{cave}}(\mathbf{x})$. This framework has recently been used to solve a wide variety of machine learning tasks, such as learning in structured models with latent variables [32, 22], kernel methods with missing entries [23] and sparse principle component analysis [26]. In CCCP, $f$ is optimized by iteratively computing a linearization of the concave part at the current iterate $\mathbf{x}^{(i)}$ and solving the resulting convex problem

$$\mathbf{x}^{(i+1)} = \arg\min_{\mathbf{x}} f_{\text{vex}}(\mathbf{x}) + \mathbf{x}^T \nabla f_{\text{cave}}(\mathbf{x}^{(i)}). \tag{2}$$

This process is guaranteed to monotonically decrease the objective and it converges globally, *i.e.*, for any point $\mathbf{x}$ (see Theorem 2 of [33] and Theorem 8 [25]). Moreover, Salakhutdinov *et al.* [19] showed that the convergence rate of CCCP, which is between super-linear and linear, depends on the curvature ratio between the convex and concave part. In order to take advantage of CCCP to solve our problem, we need to decompose the energy function into a sum of convex and concave parts. In the next section we show that this decomposition always exists. Furthermore, we provide a procedure to perform this decomposition given general polynomials.

## 2.3 Existence of a Concave-Convex Decomposition of Polynomials

Theorem 1 in [33] shows that for all arbitrary continuous functions with bounded Hessian a decomposition into convex and concave parts exists. However, Hessians of polynomial functions are not bounded in $\mathbb{R}^n$. Furthermore, [33] did not provide a construction for the decomposition. In this section we show that for polynomials this decomposition always exists and we provide a construction. Note that since odd degree polynomials are unbounded from below, *i.e.*, not proper, we only focus on even degree polynomials in the following. Let us therefore consider the space spanned by polynomial functions with an even degree $d$.

**Proposition 1.** *The set of polynomial functions $f(\mathbf{x}) : \mathbb{R}^n \to \mathbb{R}$ with even degree $d$, denoted $\mathcal{P}_d^n$, is a topological vector space. Furthermore, its dimension $\dim(\mathcal{P}_d^n) = \begin{pmatrix} n + d - 1 \\ d \end{pmatrix}$.*

*Proof.* (Sketch) According to the definition of vector spaces, we know that the set of polynomial functions forms a vector space over $\mathbb{R}$. We can then show that addition and multiplication over the polynomial ring $\mathcal{P}_d^n$ is continuous. Finally, $\dim(\mathcal{P}_d^n)$ is equivalent to computing a $d$-combination with repetition from $n$ elements [3]. $\square$

Next we investigate the geometric properties of convex even degree polynomials.

**Lemma 1.** *Let the set of convex polynomial functions $c(\mathbf{x}) : \mathbb{R}^n \to \mathbb{R}$ with even degree $d$ be $\mathcal{C}_d^n$. This subset of $\mathcal{P}_d^n$ is a convex cone.*

*Proof.* Given two arbitrary convex polynomial functions $f$ and $g \in \mathcal{C}_d^n$, let $h = af + bg$ with positive scalars $a, b \in \mathbb{R}^+$. $\forall \mathbf{x}, \mathbf{y} \in \mathbb{R}^n, \forall \lambda \in [0, 1]$, we have:

$$\begin{aligned} h(\lambda \mathbf{x} + (1 - \lambda)\mathbf{y}) &= af(\lambda \mathbf{x} + (1 - \lambda)\mathbf{y}) + bg(\lambda \mathbf{x} + (1 - \lambda)\mathbf{y}) \\ &\leq a(\lambda f(\mathbf{x}) + (1 - \lambda)f(\mathbf{y})) + b(\lambda h(\mathbf{x}) + (1 - \lambda)h(\mathbf{y})) \\ &= \lambda h(\mathbf{x}) + (1 - \lambda)h(\mathbf{y}). \end{aligned}$$

Therefore, $\forall f, g \in \mathcal{C}_d^n, \forall a, b \in \mathbb{R}^+$, we have $af + bg \in \mathcal{C}_d^n$, *i.e.*, $\mathcal{C}_d^n$ is a convex cone. $\square$

We now show that the eigenvalues of the Hessian of $f$ (hence the smallest one) continuously depend on $f \in \mathcal{P}_d^n$.

**Proposition 2.** *For any polynomial function $f \in \mathcal{P}_d^n$ with $d \geq 2$, the eigenvalues of its Hessian $\text{eig}(\nabla^2 f(\mathbf{x}))$ are continuous w.r.t. $f$ in the polynomial space $\mathcal{P}_d^n$.*

*Proof.* $\forall f \in \mathcal{P}_d^n$, given a basis $\{g_i\}$ of $\mathcal{P}_d^n$, we obtain the representation $f = \sum_i c_i g_i$, linear in the coefficients $c_i$. It is easy to see that $\forall f \in \mathcal{P}_d^n$, the Hessian $\nabla^2 f(\mathbf{x})$ is a polynomial matrix, linear in $c_i$, *i.e.*, $\nabla^2 f(\mathbf{x}) = \sum_i c_i \nabla^2 g_i(\mathbf{x})$. Let $M(c_1, \cdots, c_n) = \nabla^2 f(\mathbf{x}) = \sum_i c_i \nabla^2 g_i(\mathbf{x})$ define the Hessian as a function of the coefficients $(c_1, \cdots, c_n)$. The eigenvalues $\text{eig}(M(c_1, \cdots, c_n))$ are

equivalent to the root of the characteristic polynomial of $M(c_1, \cdots, c_n)$, *i.e.*, the set of solutions for $\det(M - \lambda I) = 0$. All the coefficients of the characteristic polynomial are polynomial expressions w.r.t. the entries of $M$, hence they are also polynomial w.r.t. $(c_1, \cdots, c_n)$ since each entry of $M$ is linear on $(c_1, \cdots, c_n)$. Therefore, the coefficients of the characteristic polynomial are continuously dependent on $(c_1, \cdots, c_n)$. Moreover, the root of a polynomial is continuously dependent on the coefficients of the polynomial [28]. Based on these dependencies, $\text{eig}(M(c_1, \cdots, c_n))$ are continuously dependent on $(c_1, \cdots, c_n)$, and $\text{eig}(M(c_1, \cdots, c_n))$ are continuous w.r.t. $f$ in the polynomial space $\mathcal{P}_d^n$. $\qquad\square$

The following proposition illustrates that the relative interior of the convex cone of even degree polynomials is not empty.

**Proposition 3.** *For an even degree function space $\mathcal{P}_d^n$, there exists a function $f(\mathbf{x}) \in \mathcal{P}_d^n$, such that $\forall \mathbf{x} \in \mathbb{R}^n$, the Hessian is strictly positive definite, i.e., $\nabla^2 f(\mathbf{x}) \succ 0$. Hence the relative interior of $\mathcal{C}_d^n$ is not empty.*

*Proof.* Let $f(\mathbf{x}) = \sum_i x_i^d + \sum_i x_i^2 \in \mathcal{P}_d^n$. It follows trivially that

$$\nabla^2 f(\mathbf{x}) = \text{diag}\left(\left[d(d-1)x_1^{d-2} + 2, d(d-1)x_2^{d-2} + 2, \cdots, d(d-1)x_n^{d-2} + 2\right]\right) \succ 0 \quad \forall \mathbf{x}.$$

$\qquad\square$

Given the above two propositions it follows that the dimensionality of $\mathcal{C}_d^n$ and $\mathcal{P}_d^n$ is identical.

**Lemma 2.** *The dimension of the polynomial vector space is equal to the dimension of the convex even degree polynomial cone having the same degree $d$ and the same number of variables $n$,* i.e., $\dim(\mathcal{C}_d^n) = \dim(\mathcal{P}_d^n)$.

*Proof.* According to Proposition 3, there exists a function $f \in \mathcal{P}_d^n$, with strictly positive definite Hessian, *i.e.*, $\forall \mathbf{x} \in \mathbb{R}^n, \text{eig}(\nabla^2 f(\mathbf{x})) > 0$. Consider a polynomial basis $\{g_i\}$ of $\mathcal{P}_d^n$. Consider the vector of eigenvalues $E(\hat{c}_i) = \text{eig}(\nabla^2(f(\mathbf{x}) + \hat{c}_i g_i))$. According to Proposition 2, $E(\hat{c}_i)$ is continuous w.r.t. $\hat{c}_i$, and $E(0)$ is an all-positive vector. According to the definition of continuity, there exists an $\epsilon > 0$, such that $E(\hat{c}_i) > 0, \forall \hat{c}_i \in \{c : |c| < \epsilon\}$. Hence, there exists a nonzero constant $\hat{c}_i$ such that the polynomial $f + \hat{c}_i g_i$ is also strictly convex. We can construct such a strictly convex polynomial $\forall g_i$. Therefore the polynomial set $f + \hat{c}_i g_i$ is linearly independent and hence a basis of $\mathcal{C}_d^n$. This concludes the proof. $\qquad\square$

**Lemma 3.** *The linear span of the basis of $\mathcal{C}_d^n$ is $\mathcal{P}_d^n$*

*Proof.* Suppose $\mathcal{P}_d^n$ is $N$-dimensional. According to Lemma 2, $\mathcal{C}_d^n$ is also $N$-dimensional. Denote $\{g_1, g_2, \cdots g_N\}$ a basis of $\mathcal{C}_d^n$. Assume there exists $h \in \mathcal{P}_d^n$ such that $h$ cannot be linearly represented by $\{g_1, g_2, \cdots g_N\}$. We have $\{g_1, g_2, \cdots, g_N, h\}$ are $N+1$ linear independent vectors in $\mathcal{P}_d^n$, which is in contradiction with $\mathcal{P}_d^n$ being $N$-dimensional. $\qquad\square$

**Theorem 1.** *$\forall f \in \mathcal{P}_d^n$, there exist convex polynomials $h, g \in \mathcal{C}_d^n$ such that $f = h - g$.*

*Proof.* Let the basis of $\mathcal{C}_d^n$ be $\{g_1, g_2, \cdots, g_N\}$. According to Lemma 3, there exist coefficients $c_1, \cdots, c_N$, such that $f = c_1 g_1 + c_2 g_2 + \cdots + c_N g_N$. We can partition the coefficients into two sets, according to their sign, *i.e.*, $f = \sum_{c_i \geq 0} c_i g_i + \sum_{c_j < 0} c_j g_j$. Let $h = \sum_{c_i \geq 0} c_i g_i$ and $g = -\sum_{c_j < 0} c_j g_j$. We have $f = h - g$, while both $h$ and $g$ are convex polynomials. $\qquad\square$

According to Theorem 1 there exists a concave-convex decomposition given any polynomial, where both the convex and concave parts are also polynomials with degree no greater than the original polynomial. As long as we can find $\begin{pmatrix} n + d - 1 \\ d \end{pmatrix}$ linearly independent convex polynomial basis functions for any arbitrary polynomial function $f \in \mathcal{P}_d^n$, we obtain a valid decomposition by looking at the sign of the coefficients. It is however worth noting that the concave-convex decomposition is not unique. In fact, there is an infinite number of decompositions, trivially seen by adding and subtracting an arbitrary convex polynomial to an existing decomposition.

Finding a convex basis is however not an easy task, mainly due to the difficulties on checking convexity and the exponentially increasing dimension. Recently, Ahmadi *et al.* [1] proved that even deciding on the convexity of quartic polynomials is NP-hard.

**Algorithm 1** CCCP Inference on Continuous MRFs with Polynomial Potentials

---
**Input:** Initial estimation $\mathbf{x}_0$
   $\forall r$ find $f_r(\mathbf{x}_r) = f_{r,\text{vex}}(\mathbf{x}_r) + f_{r,\text{cave}}(\mathbf{x}_r)$ via Eq. (4) or via a polynomial basis (Theorem 1)
   **repeat**
      solve $\mathbf{x}^{(i+1)} = \arg\min_{\mathbf{x}} \sum_r f_{r,\text{vex}}(\mathrm{x}_r) + \mathbf{x}^T \nabla_{\mathbf{x}}(\sum_{r \in \mathcal{R}} f_{r,\text{cave}}(\mathrm{x}_r^{(i)}))$ with L-BFGS.
   **until** convergence
**Output:** $\mathbf{x}_*$

---

## 2.4 Constructing a Concave-Convex Decomposition of Polynomials

In this section we derive an algorithm to construct the concave-convex decomposition of arbitrary polynomials. Our algorithm first constructs the convex basis of the polynomial vector space $\mathcal{P}_d^n$ before extracting a convex polynomial containing the target polynomial via a sum-of-squares (SOS) program. More formally, given a non-convex polynomial $f(\mathbf{x})$ we are interested in constructing a convex function $h(\mathbf{x}) = f(\mathbf{x}) + \sum_i c_i g_i(\mathbf{x})$, with $g_i(\mathbf{x})$, $i = \{1, \ldots, m\}$, the set of all convex monomials with degree no grater than $\deg(f(\mathbf{x}))$. From this it follows that $f_{\text{vex}} = h(\mathbf{x})$ and $f_{\text{cave}} = -\sum_i c_i g_i(\mathbf{x})$. In particular, we want a convex function $h(\mathbf{x})$, with coefficients $c_i$ as small as possible:

$$\min_{\mathbf{c}} \ \mathbf{w}^T \mathbf{c} \quad \text{s.t.} \quad \nabla^2 f(\mathbf{x}) + \sum_i c_i \nabla^2 g_i(\mathbf{x}) \succ 0 \quad \forall \mathbf{x} \in \mathbb{R}^n, \tag{3}$$

with the objective function being a weighted sum of coefficients. The weight vector $\mathbf{w}$ can encode preferences in the minimization, *e.g.*, smaller coefficients for larger degrees. This minimization problem is NP-hard. If it was not, we could decide whether an arbitrary polynomial $f(\mathbf{x})$ is convex by solving such a program, which contradicts the NP-hardness result of [1]. Instead, we utilize a tighter set of constraints, *i.e.*, sum-of-square constraints, which are easier to solve [14].

**Definition 2.** *For an even degree polynomial $f(\mathbf{x}) \in \mathcal{P}_d^n$, with $d = 2m$, $f$ is an SOS polynomial if and only if there exist $g_1, \ldots, g_k \in \mathcal{P}_m^n$ such that $f(\mathbf{x}) = \sum_{i=1}^{k} g_i(\mathbf{x})^2$.*

Thus, instead of solving the NP-hard program stated in Eq. (3), we optimize:

$$\min_{\mathbf{c}} \ \mathbf{w}^T \mathbf{c} \quad \text{s.t.} \quad \nabla^2 f(\mathbf{x}) + \sum_i c_i \nabla^2 g_i(\mathbf{x}) \in SOS. \tag{4}$$

The set of SOS Hessians is a subset of the positive definite Hessians [9]. Hence, every solution of this problem can be considered a valid construction. Furthermore, the sum-of-squares optimization in Eq. (4) can be formulated as an efficiently solvable semi-definite program (SDP) [10, 9]. It is important to note that the gap between the SOS Hessians and the positive definite Hessians increases as the degree of the polynomials grows. Hence using SOS constraints we might not find a solution, even though there exists one for the original program given in Eq. (3). In practice, SOS optimization works well for monomials and low-degree polynomials. For pairwise graphical models with arbitrary degree polynomials, as well as for graphical models of order up to four with maximum fourth order degree polynomials, we are guaranteed to find a decomposition. This is due to the fact that SOS convexity and polynomial convexity coincide (Theorem 5.2 in [2]). Most practical graphical models are within this set. Known counter-examples [2] are typically found using specific tools.

We summarize our algorithm in Alg. 1. Given a graphical model with polynomial potentials with degree at most $d$, we obtain a concave-convex decomposition by solving Eq. (4). This can be done for the full polynomial or for each non-convex monomial. We then apply CCCP in order to perform inference, where we solve a convex problem at each iteration. In particular, we employ L-BFGS, mainly due to its super-linear convergence and its storage efficiency [12]. In each L-BFGS step, we apply a line search scheme based on the Wolfe conditions [12].

## 2.5 Extensions

**Dealing with very large graphs:** Motivated by recent progress on accelerating graphical model inference [7, 21, 20], we can handle large-scale problems by employing dual decomposition and using our approach to solve the sub-problems.

**Non-polynomial cases:** We have described our method in the context of graphical models with polynomial potentials. It can be extended to the non-polynomial case if the involved functions have

|               | L-BFGS  | PCBP   | FusionMove | ADMM-Poly | Ours         |
| ------------- | ------- | ------ | ---------- | --------- | ------------ |
| Energy        | 10736.4 | 6082.7 | 4317.7     | 3221.1    | 3062.8       |
| RMSE (mm)     | 4.98    | 4.50   | 2.95       | 3.82      | 3.07         |
| Time (second) | 0.11    | 56.60  | 0.12       | 18.32     | 8.70 (×2)    |

Table 1: 3D Reconstruction on $3 \times 3$ meshes with noise variance $\sigma = 2$.

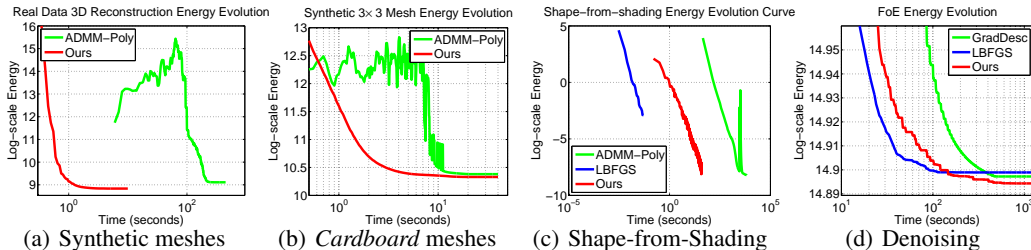

| (a) Synthetic meshes | (b) *Cardboard* meshes | (c) Shape-from-Shading | (d) Denoising |

Figure 1: Average energy evolution curve for different applications.

bounded Hessians, since we can still construct the concave-convex decomposition. For instance, for the Lorentzian regularizer $\rho(x) = \log(1 + \frac{x^2}{2})$, we note that $\rho(x) = \{\log(1 + \frac{x^2}{2}) + \frac{x^2}{8}\} - \frac{x^2}{8}$ is a valid concave-convex decomposition. We refer the reader to the supplementary material for a detailed proof. Alternatively, we can approximate any continuous function with polynomials by employing a Taylor expansion around the current iterate, and updating the solution via one CCCP step within a trust region.

# 3 Experimental Evaluation

We demonstrate the effectiveness of our approach using three different applications: non-rigid 3D reconstruction, shape from shading and image denoising. We refer the reader to the supplementary material for more figures as well as an additional toy experiment on a densely connected graph with box constraints.

## 3.1 Non-rigid 3D Reconstruction

We tackle the problem of deformable surface reconstruction from a single image. Following [30], we parameterize the 3D shape via the depth of keypoints. Let $\mathbf{x} \in \mathbb{R}^N$ be the depth of $N$ points. We follow the locally isometric deformation assumption [20], *i.e.*, the distance between neighboring keypoints remains constant as the non-rigid surface deforms. The 3D reconstruction problem is then formulated as

$$\min_{\mathbf{x}} \sum_{(i,j) \in \mathcal{N}} \left( \|x_i \mathbf{q}_i - x_j \mathbf{q}_j\|^2 - d_{i,j}^2 \right)^2, \tag{5}$$

where $d_{i,j}$ is the distance between keypoints (given as input), $\mathcal{N}$ is the set of all neighboring pixels, $x_i$ is the unknown depth of point $i$, $\mathbf{q}_i = \mathbf{A}^{-1}(u_i, v_i, 1)^T$ is the line-of-sight of pixel $i$ with $\mathbf{A}$ denoting the known internal camera parameters. We consider a six-neighborhod system, *i.e.*, up, down, left, right, upper-left and lower-right. Note that each pairwise potential is a four-degree non-convex polynomial with two random variables. We can easily decompose it into 15 monomials, and perform a concave-convex decomposition given the corresponding convex polynomials (see supplementary material for an example).

We first conduct reconstruction experiments on the 100 randomly generated $3 \times 3$ meshes of [20], where zero-mean Gaussian noise with standard deviation $\sigma = 2$ is added to each observed keypoint coordinate. We compare our approach to Fusion Moves [30], particle convex belief propagation (PCBP) [17], L-BFGS as well as dual decomposition with the alternating direction method of multipliers using a polynomial solver (ADMM-Poly) [20]. We employ three different metrics, energy at convergence, running time and root mean square error (RMSE). For L-BFGS and our method, we use a flat mesh as initialization with two rotation angles $(0, 0, 0)$ and $(\pi/4, 0, 0)$. The convergence criteria is an energy decrease of less than $10^{-5}$ or a maximum of 500 iterations is reached. As shown in Table 1 our algorithm achieves lower energy, lower RMSE, and faster running time than ADMM-Poly and PCBP. Furthermore, as shown in Fig. 1(a) the time for running our algorithm to convergence is similar to a single iteration of ADMM-Poly, while we achieve much lower energy.

|             | L-BFGS | CLVM | ADMM-Poly | Ours   |
|-------------|--------|------|-----------|--------|
| Energy      | 736.98 | N/A  | 905.37    | 687.21 |
| RMSE (mm)   | 4.16   | 7.23 | 5.68      | 3.29   |
| Time (second) | 0.3406 | N/A | 314.8    | 10.16  |

Table 2: 3D Reconstruction on *Cardboard* sequences.

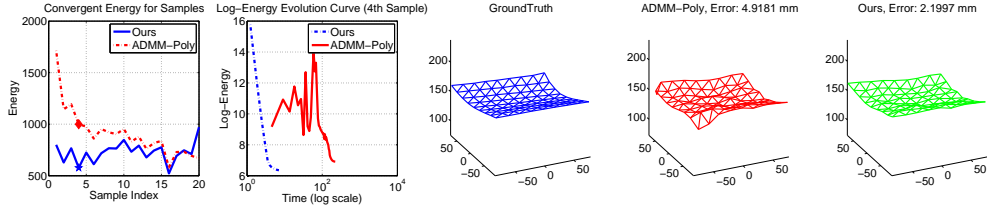

Figure 2: 3D reconstruction results on *Cardboard*. Left to right: sample comparison, energy curve, groundtruth, ADMM-Poly and our reconstruction.

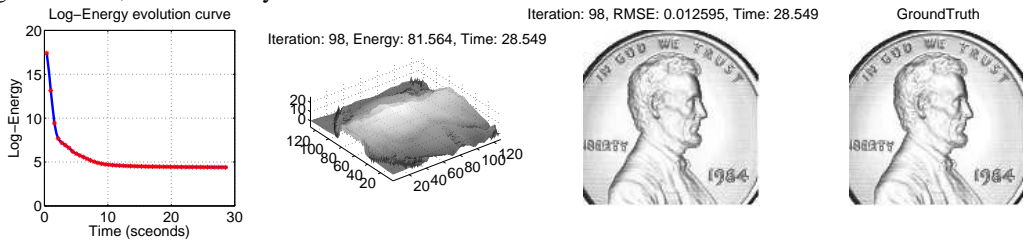

Figure 3: Shape-from-Shading results on *Penny*. Left to right: energy curve, inferred shape, rendered image with inferred shape, groundtruth image.

We next reconstruct the real-world $9 \times 9$ *Cardboard* sequence [20]. We compare with both ADMM-Poly and L-BFGS in terms of energy, time and RMSE. We also compare with the constrained latent variable model of [29], in terms of RMSE. We cannot compare the energy value since the energy function is different. Again, we use a flat mesh as initialization. As shown in Table 2, our algorithm outperforms all baselines. Furthermore, it is more than 20 times faster than ADMM-Poly, which is the second best algorithm. Average energy as a function of time is shown in Fig. 1(b). We refer the reader to Fig. 2 and the video in the supplementary material for a visual comparison between ADMM-Poly and our method. From the first subfigure we observe that our method achieves lower energy for most samples. The second subfigure illustrate the fact that our approach monotonically decreases the energy, as well as our method being much faster than ADMM-Poly.

### 3.2 Shape-from-Shading

Following [5, 20], we formulate the shape from shading problem with 3rd-order 4-th degree polynomial functions. Let $\mathbf{x}_{i,j} = (u_{i,j}, v_{i,j}, w_{i,j})^T$ be the 3D coordinates of each triangle vertex. Under the Lambertian model assumption, the intensity of a triangle $r$ is represented as: $I_r = \frac{l_1 p_r + l_2 q_r + l_3}{\sqrt{p_r^2 + q_r^2 + 1}}$, where $\mathbf{l} = (l_1, l_2, l_3)^T$ is the direction of the light, $p_r$ and $q_r$ are the $x$ and $y$ coordinates of normal vector $\mathbf{n}_r = (p_r, q_r, 1)^T$, which is computed as $p_r = \frac{(v_{i,j+1}-v_{i,j})(w_{i+1,j}-w_{i,j})-(v_{i+1,j}-v_{i,j})(w_{i,j+1}-w_{i,j})}{(u_{i,j+1}-u_{i,j})(v_{i+1,j}-v_{i,j})-(u_{i+1,j}-u_{i,j})(v_{i,j+1}-v_{i,j})}$ and $p_r = \frac{(u_{i,j+1}-u_{i,j})(w_{i+1,j}-w_{i,j})-(u_{i+1,j}-u_{i,j})(w_{i,j+1}-w_{i,j})}{(u_{i,j+1}-u_{i,j})(v_{i+1,j}-v_{i,j})-(u_{i+1,j}-u_{i,j})(v_{i,j+1}-v_{i,j})}$, respectively. Each clique $r$ represents a triangle, which is constructed by three neighboring points on the grid, *i.e.*, either $(\mathbf{x}_{i,j}, \mathbf{x}_{i,j+1}, \mathbf{x}_{i+1,j})$ or $(\mathbf{x}_{i,j}, \mathbf{x}_{i,j-1}, \mathbf{x}_{i+1,j})$. Given the rendered image and lighting direction, shape from shading is formulated as

$$\min_{\mathbf{w}} \sum_{r \in \mathcal{R}} \left( (p_r^2 + q_r^2 + 1)I_r^2 - (l_1 p_r + l_2 q_r + l_3)^2 \right)^2 . \tag{6}$$

We tested our algorithm on the *Vase*, *Penny* and *Mozart* datasets, where *Vase* and *Penny* are $128 \times 128$ images and *Mozart* is a $256 \times 256$ image with light direction $\mathbf{l} = (0, 0, 1)^T$. The energy evolution curve, the inferred shape as well as the rendered and ground-truth images are illustrated in Fig. 3. See the supplementary material for more figures on *Penny* and *Mozart*. Our algorithm achieves very low energy, producing very accurate results in only 30 seconds. ADMM-Poly hardly runs on such large-scale data due to the computational cost of the polynomial system solver (more than 2 hours

|  | L-BFGS | GradDesc | Ours |
|---|---|---|---|
| Energy | 29547 | 29598 | 29413 |
| PSNR | 30.96 | 31.56 | 31.43 |
| Time (sec) | 189.5 | 1122.5 | 384.5 |

Table 3: FoE Energy Minimization Results.

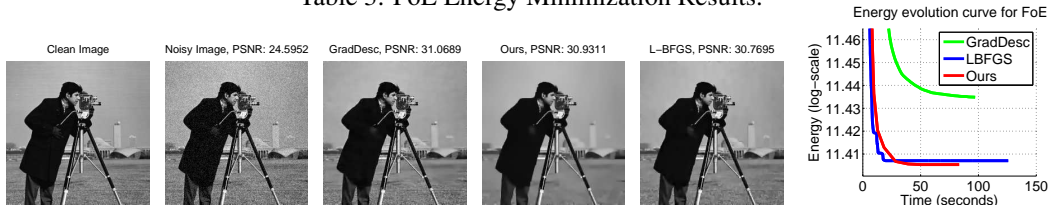

Figure 4: FoE based image denoising results on *Cameraman*, $\sigma = 15$.

per iteration). In order to compare with ADMM-Poly, we also conduct the shape from shading experiment on a scaled $16 \times 16$ version of the *Vase* data. Both methods retrieve a shape that is very close to the global optimum (0.00027 for ADMM-Poly and 0.00032 for our approach), however, our algorithm is over 500 times faster than ADMM-Poly (2250 seconds for ADMM-Poly and 13.29 seconds for our proposed method). The energy evolution curve on the $16 \times 16$ re-scaled image in shown in Fig. 1(c).

### 3.3 Image Denoising

We formulate image denoising via minimizing the Fields-of-Experts (FoE) energy [18]. The data term encodes the fact that the recovered image should be close to the noisy input, where closeness is weighted by the noise level $\sigma$. Given a pre-learned linear filterbank of 'experts' $\{\mathbf{J}_i\}_{i=1,\ldots,K}$, the image prior term encodes the fact that natural images are Gibbs distributed via $p(\mathbf{x}) = \frac{1}{Z}\exp(\prod_{r\in\mathcal{R}}\prod_{i=1}^{K}(1+\frac{1}{2}(\mathbf{J}_i^T\mathbf{x}_r)^2)^{\alpha_i})$. Thus we formulate denoising as

$$\min_{\mathbf{x}} \frac{\lambda}{\sigma^2}\|\mathbf{x}-\mathbf{y}\|_2^2 + \sum_{r\in\mathcal{R}}\sum_{i=1}^{K}\alpha_i\log(1+\frac{1}{2}(\mathbf{J}_i^T\mathbf{x}_r)^2), \tag{7}$$

where $\mathbf{y}$ is the noisy image input, $\mathbf{x}$ is the clean image estimation, $r$ indexes $5 \times 5$ cliques and $i$ is the index for each FoE filter. Note that this energy function is not a polynomial function. However, for each FoE model, the Hessian of the energy function $\log(1+\frac{1}{2}(\mathbf{J}_i^T\mathbf{x}_r)^2)$ is lower bounded by $-\frac{J_i^T J_i}{8}$ (proof in the supplementary material). Therefore, we simply add an extra term $\gamma\mathbf{x}_r^T\mathbf{x}_r$ with $\gamma > \frac{J_i^T J_i}{8}$ to obtain the concave-convex decomposition $\log(1+\frac{1}{2}(\mathbf{J}_i^T\mathbf{x}_r)^2) = \{\log(1+\frac{1}{2}(\mathbf{J}_i^T\mathbf{x}_r)^2)+\gamma\mathbf{x}_r^T\mathbf{x}_r\} - \gamma\mathbf{x}_r^T\mathbf{x}_r$. We utilize a pre-trained $5 \times 5$ filterbank with 24 filters, and conduct experiments on the BM3D benchmark [1] with noise level $\sigma = 15$. In addition to the other baselines, we compare to the original FoE inference algorithm, which essentially is a first-order gradient descent method with fixed gradient step [18]. For L-BFGS, we set the maximum number of iterations to 10,000, to make sure that the algorithm converges. As shown in Table 3 and Fig. 1(d), our algorithm achieves lower energy than L-BFGS and first-order gradient descent. Furthermore, we see that lower energy does not translate to higher PSNR, showing the limitation of FoE as an image prior.

## 4 Conclusions

We investigated the properties of polynomials, and proved that every multivariate polynomial with even degree can be decomposed into a sum of convex and concave polynomials with degree no greater than the original one. Motivated by this property, we exploited the concave-convex procedure to perform inference on continuous Markov random fields with polynomial potentials. Our algorithm is especially fit for solving inference problems on continuous graphical models, with a large number of variables. Experiments on non-rigid reconstruction, shape-from-shading and image denoising validate the effectiveness of our approach. We plan to investigate continuous inference with arbitrary differentiable functions, by making use of polynomial approximations as well as tighter concave-convex decompositions.

## Footnotes

[1] http://www.cs.tut.fi/~foi/GCF-BM3D/

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
