[Supplementary Material]

# Supplementary Material: Efficient Inference of Continuous Markov Random Fields with Polynomial Potentials

**Shenlong Wang**
University of Toronto
slwang@cs.toronto.edu

**Alexander G. Schwing**
University of Toronto
aschwing@cs.toronto.edu

**Raquel Urtasun**
University of Toronto
urtasun@cs.toronto.edu

In this supplementary material we first show an additional experiment for a densely connected graph using a toy denoising example. After that, we provide a convex polynomial basis example on $\mathcal{P}_4^2$, and illustrate how to use this basis to perform a concave-convex decomposition of the Rosenbrock function. In addition, we give a proof on how to construct a concave-convex decomposition for the Fields-of-Expert (FoE) models.

## 1   Experiments on Densely Connected Graphs

One advantage of our proposed algorithm is that it is efficient on densely (or even fully) connected graphs. Furthermore, we can employ tools for constrained convex optimization to integrate restrictions on the feasible set. We further validate the efficiency of our approach using a simple example with a densely connected graph. We are given a noisy image and a binary segmentation with two different labels. Our task is to denoise the image by exploiting the fact that the noise of the two segments might be different. To make the task more challenging we observe only a small fraction of the labeling relationships between pixels, *i.e.*, whether the labels of two pixels take an identical or the opposite label. We model the problem as follows:

$$\arg\min_{\mathbf{x}} \frac{1}{\sigma^2} \|\mathbf{x} - \mathbf{y}\|_2^2 + \lambda \left( \sum_{\mathcal{L}_i = \mathcal{L}_j, (i,j) \in \mathbf{O}} \|x_i - x_j\|_2^2 - \sum_{\mathcal{L}_i \neq \mathcal{L}_j, (i,j) \in \mathbf{O}} \|x_i - x_j\|_2^2 \right) \quad (1)$$

$$\text{s. t. } 0 \leq x_i \leq 255, \forall i, \quad (2)$$

where $\mathcal{L}_i \in \{0, 1\}$ is the label of a certain pixel and $\mathbf{O}$ is the set of pixel pairs for which the labeling relationship is observed. Note that the maximum number of potentials is $\frac{n^2}{2}$, which is very large for real-world applications. For example, there are $2 \cdot 10^{11}$ potentials for a $1024 \times 768$ image. ADMM-Poly and PCBP cannot perform inference on such a densely connected graphical model. However, our approach can handle these cases. To our advantage is the fact that this problem is a convex-concave function by nature. Hence we can use our proposed algorithm to tackle this formulation. Note that unlike previous experiments, this energy minimization problem has a box constraint for each pixel. Therefore, we need to adapt L-BFGS in order to tackle the convex sub-problem. Alternatively, since this is a quadratic programming problem, we exploit Gurobi [1] to solve the sub-problem in each iteration. We tried 12 different percentages varying from $100\%$ to $0.01\%$ for an image of size $35 \times 84$ with zero-mean Gaussian noise with variance $\sigma = 100$. The results illustrated in Fig. 1 show that our approach achieves good results with limited number of observations, and in a reasonable running time for fully connected graph ($100\%$). Fig. 2 provides a visual comparison.

Figure 1: Densely Connected Labeled Image Denoising

Figure 2: An Visual Comparison on Densely Connected Graph Denoising

## 2 Concave-Convex Decomposition on $\mathcal{P}_4^2$

Next we illustrate how to construct a convex basis for $\mathcal{P}_d^n$ using $\mathcal{P}_4^2$ as an example. According to Proposition 1 in our submission, we know that $\dim(\mathcal{P}_4^2) = \sum_{d=0}^{4} \binom{2+d-1}{d} = 15$. The basis can be constructed using the following 15 monomials which are shown on the left hand side of Eq. (3). For each convex monomial we seek the tightest convex representation by adding a convex polynomial. 'The tightest' means we can not subtract any tiny convex part to make the representation still convex. The right hand side in Eq. (3) shows the convex polynomial basis that we obtain.

$$
\begin{array}{cc}
\text{monomial} & \text{convex polynomial basis} \\
x_1 & x_1 \\
x_2 & x_2 \\
x_1^2 & x_1^2 \\
x_1 x_2 & \frac{1}{2}x_1^2 + x_1 x_2 + \frac{1}{2}x_2^2 \\
x_2^2 & x_2^2 \\
x_1^3 & \frac{1}{12}x_1^4 + x_1^3 + \frac{9}{2}x_1^2 \\
x_1^2 x_2 & \frac{1}{2}x_1^4 + \frac{1}{2}x_2^4 + x_1^2 x_2 + x_1^2 + x_2^2 \\
x_1 x_2^2 & \frac{1}{2}x_1^4 + \frac{1}{2}x_2^4 + x_1 x_2^2 + x_1^2 + x_2^2 \\
x_2^3 & \frac{1}{12}x_2^4 + x_2^3 + \frac{9}{2}x_2^2 \\
x_1^4 & x_1^4 \\
x_1^3 x_2 & \frac{4}{5}x_1^4 + x_1^3 x_2 + \frac{1}{2}x_1^2 x_2^2 + \frac{4}{5}x_2^4 \\
x_1^2 x_2^2 & \frac{1}{6}x_1^4 + x_1^2 x_2^2 + \frac{1}{6}x_2^4 \\
x_1 x_2^3 & \frac{4}{5}x_1^4 + \frac{1}{2}x_1^2 x_2^2 + x_1 x_2^3 + \frac{4}{5}x_2^4 \\
x_2^4 & x_2^4 \\
1 & 1
\end{array}
\tag{3}
$$

In the following example, we illustrate how to perform a concave-convex decomposition on a polynomial. In particular, we take the Rosenbrock function. By using the above monomials in Eq. (3) as

a lookup table, we decompose the Rosenbrock function as follows:

$$
\begin{aligned}
f &= (1-x)^2 + 100(y-x^2)^2 \\
&= 1 - 2x + x^2 + 100y^2 - 200yx^2 + x^4 \\
&= 1 - 2x + x^2 + 100y^2 + x^4 - 200\{\underbrace{xy^2 + \left(\frac{1}{2}y^4 + \frac{1}{2}x^4 + y^2 + x^2\right)}_{\text{convex polynomial basis containing } xy^2} - \left(\frac{1}{2}y^4 + \frac{1}{2}x^4 + y^2 + x^2\right)\} \\
&= \underbrace{300y^2 + 200x^4 + 100y^4 + 201x^2 - 2x + 1}_{convex} \underbrace{-(200yx^2 + 100x^4 + 100y^4 + 200x^2 + 200y^2)}_{concave} \\
&= f_{\text{vex}} + f_{\text{cave}}
\end{aligned}
$$

## 3 Concave-Convex Decomposition of the Fields-of-Expert energy

Let us consider the following function:

$$
f = \log\left(1 + \frac{1}{2}(\mathbf{J}^T\mathbf{x})^2\right) \tag{4}
$$

Its gradient and its Hessian are:

$$
\nabla f = \frac{\mathbf{J}^T\mathbf{J}\mathbf{x}}{1 + \frac{1}{2}(\mathbf{J}^T\mathbf{x})^2}, \nabla^2 f = \frac{1 - \frac{1}{2}(\mathbf{J}^T\mathbf{x})^2}{(1 + \frac{1}{2}(\mathbf{J}^T\mathbf{x})^2)^2}\mathbf{J}\mathbf{J}^T \tag{5}
$$

Moreover, since $\forall \mathbf{x} \in \mathbb{R}^n, \frac{1 - \frac{1}{2}(\mathbf{J}^T\mathbf{x})^2}{(1 + \frac{1}{2}(\mathbf{J}^T\mathbf{x})^2)^2} \geq -\frac{1}{8}$ we obtain:

$$
\text{eig}(\nabla^2 f) \geq -\frac{1}{8}\text{eig}(\mathbf{J}\mathbf{J}^T) \geq -\frac{1}{8}\mathbf{J}^T\mathbf{J} \tag{6}
$$

According to above derivation we construct the concave-convex decomposition to minimize the Fields-of-Expert energy as described in Section 3.3 of our main submission.

## Footnotes

[1] http://www.gurobi.com