[Reviews · NeurIPS 2014]

Submitted by Assigned_Reviewer_15

This is a very interesting and substantially novel paper that introduces an approach to solving continuous Markov random field energies with polynomial potentials. An insightful and well-motivated approach towards this end (ADMM-Poly) was published at CVPR 2013 [20] and is the obvious baseline to compare against. The present approach is convincingly shown to be preferable, as it is both elegant and computationally efficient.

The main idea underlying the approach is to decompose the polynomials into a difference of convex functions. Towards this end, a constructive approach is introduced for polynomials of even degree, one of the main contributions of the paper. As the authors show, the decomposition problem has a formulation as a semi-definite program and can be solved using standard solvers. Given that decomposition, the overall inference task reduces to a DC programming problem, which the authors approach using the convex-concave procedure (CCCP).

The main weakness of the approach seems to be that the semi-definite program for finding a DC decomposition of the polynomials may not have a solution, as it is only an approximation to the exact but NP-hard problem. Even though the authors mention that the approach works well for polynomials of low degree in practice (confirmed by experiments), this point is somewhat worrisome. I would encourage the authors to elaborate on this point, as it seems like a potential barrier to adoption of the approach. How often does this situation occur in practice (if at all)? For which polynomials? Are there any workarounds? Clearly, an alternative strategy is needed in case this situation occurs.

As the authors show, in some cases, for some continuous energies, a DC decomposition is apparent even without having to resort to the above construction algorithm. The approach then basically becomes an application of CCCP to continuous energies.

Quality:
The paper is technically sound, though, as mentioned above, a more in-depth discussion of the potential failure to construct the decomposition of the polynomials would be appreciated. The experiments are impressively diverse and the results are very convincing. The proposed approach clearly works better than ADMM-Poly, the main baseline to compare against. It is faster and often obtains solutions of better quality.

Clarity:
The paper is well-written, though not always easy to read, owing to the technical depth of the material. One aspect that the authors may wish to clarify is the claimed global convergence of their approach (inherited from CCCP). I assume that this is meant to be interpreted as "the algorithm converges, for *any* initialization", rather than "the algorithm converges to a global optimum". As far as I am aware, the convergence properties of CCCP only guarantee convergence to either an optimum or a saddle point. As it stands, readers might misinterpret the manuscript to assume that the proposed approach always finds a global optimum.
Along the same line, the authors claim in lines 039ff that Gaussian belief propagation "retrieves the optimum for arbitrary graphs when the potentials are Gaussian". This is not true. There are two known convergence conditions relating to the diagonal dominance, and the spectral radius, of the system matrix, respectively.

Originality:
The proposed approach is novel to this reviewer, and substantial theory is developed along the way.

Significance:
The proposed approach has broad and important applications in computer vision and imaging. The experimental results are impressive, and the approach hence has the potential to significantly benefit many members of these communities.
Summary: In summary, this is a substantially novel submission that develops new theory and demonstrates strong experimental results. For these reasons, I recommend that the paper be accepted.

Submitted by Assigned_Reviewer_23

The work shows how the energy of a continuous Markov Random Field that consists
of polynomial pairwise energies can be decomposed into a sum of a convex and
concave polynomial. This leverages the use of the concave-convex procedure
(CCCP) to do fast MAP inference.

Quality: The paper states valid proofs for the used methodology and it seems
reproducible. The diverse set of experiments further show that the proposed
method performs well. In comparison to other optimization techniques the
polynomial decomposition usually finds better solutions. Nevertheless, the
paper could be more clear about the weaknesses. E.g. the trade-off between
wall-clock time for solving the optimization problem and sometimes only
marginally better solution could be discussed.

Clarity: The paper is well written and the presentation is clear. I liked the
pace at which the work introduces the splitting of the polynomial into the two
parts of interest.

Originality: Using the decomposition of a polynomial to drive inference of a
MRF seems to be new.

Significance: MRFs with polynomial energies are certainly of interest, e.g. in
Computer Vision.

Strengths:
+ This is certainly an important problem in e.g. Computer Vision and worth
studying.
+ Nice presentation; clear proofs.
+ Diverse experiments.

Weaknesses:
- The authors could be a bit more clear about the weaknesses of their work: The
method seems to yield better solutions for the resulting optimization problem
but seems to needs more wall-clock time than the competing direct
optimization with L-BFGS.

Questions to the authors:
- Do the reported running times include the solving time for the decomposition
into convex and concave functions?
Summary: The work addresses a relevant problem and proposes an interesting solution.
Proofs and exposition are clear. The paper could state the weaknesses more
clearly: The method seems to be slower for `real-world' problems than the
competitors but yields better solutions.

Submitted by Assigned_Reviewer_36

In this paper, the authors first show that the multivariate polynomial functions with even degree can be decomposed into convex and concave parts. They use this property to solve MAP inference problem in continuous Markov Random Fields. Finally they show experiments on three problems: 3D reconstruction, shape from shading and image denoising.

--> The paper is well written and tackle very important problem of solving continuous MRF problems. They provide theoretical analysis of their algorithm.

Minor comments:

--> The authors claim to significantly outperform existing techniques in terms of efficiency and accuracy. However, this is not completely reflected in their result section. For example in all the three cases, their algorithm takes longer time than L-BFGS approach. If this is the case then they should clearly mention in the text.

--> Authors should explain why they observe up and down in their energy versus sample index curve in Fig. 2.

--> I would suggest to properly explain the tables in their captions.
Summary: The paper provides a theoretically sound algorithm for decomposing an even degree polynomial into a convex and concave parts. They properly analyse the properties of the algorithm and use it to solve continuous MRF problems. The paper is well written.
Author Feedback
Author rebuttal: We thank the reviewers for their helpful and positive feedback. We will incorporate all their comments in a revised version of the manuscript.

To R15 w.r.t finding a decomposition: We note that, for all pairwise graphical models with arbitrary degree polynomials, as well as for graphical models of order up to four with maximum fourth order degree polynomials, we are guaranteed to find a decomposition. This is due to the fact that SOS convexity and polynomial convexity coincide in those cases(see Theorem 5.2 in reference [A] below). Most practical graphical models are within this set. Remarkably, we have not yet encountered any convex but not SOS-convex polynomial in practice. Known counter-examples [A] are typically found using specific tools. We will add a discussion about this in the paper.

To R15: We agree with the reviewer and we will make our statement about Gaussian BP more precise.

To R15: By ‘globally convergent’ we mean convergence for arbitrary initialization to a local optima/saddle point. We will clarify this in the paper.

To R23 w.r.t. reproducibility: We will make our code available to reproduce all experiments.

To R23 and R36: We expect our approach to be slower than L-BFGS as we employ L-BFGS to solve the intermediate convex optimization steps obtained through CCCP. The experiment is rather intended to illustrate that a standard solver does indeed converge to a worse local optima than our approach. We’ll clarify this in a revised version.

To R23: Reported times do not include concave-convex decomposition but for FOE experiments, the decomposition is trivially obtainable in no time by adding unary quadratic terms with constant coefficients. Decomposition of the other two applications takes about 0.07s for 9x9 Cardboard and 8.47s for 128x128 vase image. This is only a small fraction of the reported time and we’ll clarify this fact. It is important to note that construction of the monomial basis is independent of the considered application and can be pre-computed.

To R36: Fig 2 depicts the achieved energy as a function of the example index (i.e., image index). As some examples are more difficult than others we do not expect any particular shape for this plot. We will clarify this.

New Reference:
[A] A. Ahmadi, P. Parrilo. A complete characterization of the gap between convexity and SOS-convexity. SIAM Journal on Optimization, 2013